# Citral Modulates MMP-2 and MMP-9 Activities on Healing of Gastric Ulcers Associated with High-Fat Diet-Induced Obesity

**DOI:** 10.3390/ijms24054888

**Published:** 2023-03-03

**Authors:** Rie Ohara, Felipe Lima Dario, Maycon Tavares Emílio-Silva, Renata Assunção, Vinícius Peixoto Rodrigues, Gabriela Bueno, Priscila Romano Raimundo, Lúcia Regina Machado da Rocha, Clelia Akiko Hiruma-Lima

**Affiliations:** Department of Structural and Functional Biology (Physiology), Institute of Biosciences of Botucatu, São Paulo State University, Botucatu 18618-970, SP, Brazil

**Keywords:** gastric ulcer, healing, obesity, citral, monoterpene, matrix metalloproteinases

## Abstract

Obesity causes low-grade inflammation that results in the development of comorbidities. In people with obesity, exacerbation of gastric lesion severity and delayed healing may aggravate gastric mucosal lesions. Accordingly, we aimed to evaluate the citral effects on gastric lesion healing in eutrophic and obese animals. C57Bl/6 male mice were divided into two groups: animals fed a standard diet (SD) or high-fat diet (HFD) for 12 weeks. Gastric ulcers were induced using acetic acid (80%) in both groups. Citral (25, 100, or 300 mg/kg) was administered orally for 3 or 10 days. A vehicle-treated negative control (1% Tween 80, 10 mL/kg) and lansoprazole-treated (30 mg/kg) were also established. Lesions were macroscopically examined by quantifying regenerated tissue and ulcer areas. Matrix metalloproteinases (MMP-2 and -9) were analyzed by zymography. The ulcer base area between the two examined periods was significantly reduced in HFD 100 and 300 mg/kg citral-treated animals. In the 100 mg/kg citral-treated group, healing progression was accompanied by reduced MMP-9 activity. Accordingly, HFD could alter MMP-9 activity, delaying the initial healing phase. Although macroscopic changes were undetectable, 10-day treatment with 100 mg/kg citral exhibited improved scar tissue progression in obese animals, with reduced MMP-9 activity and modulation of MMP-2 activation.

## 1. Introduction

In 2016, it was estimated that 39% of the global adult population aged ≥ 18 years (39% of males and 40% of females) were overweight, and 13% were deemed obese [1]. Obesity is characterized by excessive adipose tissue accumulation due to an imbalance between high consumption and low energy expenditure [2]. The excessive adipose tissue is deposited as visceral adipose tissue, which is well-known to secrete pro-inflammatory cytokines, resulting in a low-grade inflammatory condition. More recently, visceral adipose tissue accumulation has been associated with the development of comorbidities, such as insulin resistance, dyslipidemia, hypertension, and cardiovascular disease [3]. Obesity-induced systemic inflammation is triggered by immune cell recruitment, the interaction and activation of these immune cells, and the release of inflammatory molecules. In addition, a combination of other events participates in maintaining this inflammatory condition [4]. The offer of a high-fat diet (HFD) is an important preclinical model for the obesity study in animals, as it induces an increase in adipose tissue and has been identified as an important factor in the inflammation development during the disease [5]. Rapid adipose tissue expansion after ingestion of an HFD can induce systemic physiological changes that directly impact the functioning of certain organs [4]. Individuals with obesity often experience altered gastric emptying time [6]. Abdominal obesity, known to be characterized by increased visceral fat, has been associated with the onset of several gastrointestinal tract diseases, owing to an increase in intra-abdominal pressure due to excessive adipose tissue accumulation [7]. Furthermore, adipose tissue can locally secrete adipocytokines and other factors, such as tumor necrosis factor (TNF)-α and interleukin (IL)-6, which may be involved in the aggravate gastric lesions; this has been observed in studies investigating the mechanism of action of antiulcerogenic agents, where the inhibition of these cytokines afforded gastroprotection [7,8].

The association between obesity and peptic ulcers has been previously explored; however, the results remain controversial. Recently, a cohort study demonstrated a direct correlation between genetically predicted obesity and peptic ulcer disease, mainly, when observed with non-NSAIDs user patients, another important factor of the development of peptic ulcers [9]. Furthermore, it can be speculated that mechanisms underlying the potential association between obesity and peptic ulcers could be aggravated by low-grade chronic inflammation [7]. Although an association between the two diseases remains elusive, obesity-induced poor vascularization directly impacts the skin healing process, given the hindered recruitment of angiogenic factors necessary for the complete healing of the lesion [10]. Disruption of the mucous layer of the stomach results in ulcer formation. Several factors have been implicated in ulcer formation, including decreased production of mucus and bicarbonate, which may be attributed to the insufficient blood supply, Helicobacter pylori infection, or overuse of nonsteroidal anti-inflammatory drugs (NSAIDs) [11,12] Gastric lesions can be histologically classified into two parts: the base and margin of the ulcer. The base of the ulcer is the necrotic region, where granulation connective tissue, comprising fibroblasts, macrophages, and endothelial cells, is known to predominate. Conversely, the ulcer margin, referred to as regenerating tissue in the present study, consists of dedifferentiated proliferating epithelial tissue, expressing high levels of growth factors [13]. 

Currently, the main treatment strategy for gastric ulcers involves the use of acid secretion inhibitors, primarily proton pump inhibitors and histamine receptor antagonists; however, these drugs often induce adverse reactions, such as abdominal pain, nausea, constipation, and flatulence. Prolonged use of proton pump inhibitors has also been associated with liver damage and an increased risk of developing gastric cancer, thereby restricting the treatment duration, in addition to causing adverse effects in other systems, such as kidney disease, dementia, and bone fractures [14,15]. Therefore, all currently employed drugs fail to promote effective re-epithelialization of the lesion with new vessel formation for adequate blood supply to the tissue, thereby leading to ulcer recurrence and worsening [16]. Similar to gastric ulcer formation, the healing process is multifactorial and complex and involves re-epithelialization, restoration of glands, angiogenesis, and extracellular matrix (ECM) deposition [13]. 

The ECM is a complex multimolecular structure comprising collagen and elastin fibers and structural glycoproteins, including fibronectin, laminin, and mucopolysaccharides. Under physiological conditions, a balance exists between the synthesis, deposition, and degradation of ECM components, and its composition varies among multicellular structures, with fibroblasts and epithelial cells being the most common cell types [17]. The main enzymes participating in ECM synthesis and degradation include matrix metalloproteinases (MMPs), zinc-dependent proteases that play a crucial role in ECM remodeling via proteolytic degradation of its components, surface protein activation, and release of membrane-bound receptor molecules [18]. MMPs with gelatinase activity hydrolyze gelatin into polypeptides, peptides, and amino acids, which are subsequently secreted across the cell membrane. MMP-2 and -9 are gelatinases that facilitate the binding of gelatin and collagen through three fibronectin type II-like repeat domains inserted into the catalytic domain of the structure. MMP-2 and -9 are fundamental in the healing process, as they play a pivotal role in accelerating cell migration and re-epithelialization, respectively [18,19]. In gastric ulcers, degradation of the gastric mucosa is directly related to ECM degradation. MMP-9 is secreted mainly by neutrophils and macrophages and acts during the initial phase of healing. Patients with gastric ulcers reportedly exhibit increased MMP-9 production at the edge of the lesion. Furthermore, elevated MMP-9 activity was detected in tissues where the lesion was associated with a high risk of ulcer recurrence, suggesting that MMP-9 is a marker for poor healing [20]. MMP-2 is primarily secreted by fibroblasts and leukocytes, and its activity is considered critical during the initial phase because it accelerates cell migration. However, in the proliferative phase, this increase is related to the fragility of the regenerated tissue [21]. Some pathological processes, such as adipose tissue expansion, are involved in regulating proteolytic enzymes. Reportedly, patients with obesity exhibit increased plasma levels of MMP-2 and -9. In addition, HFD-fed animals exhibited increased MMP-9 expression in visceral adipose tissues [17,22]. 

Considering the multifactorial nature of obesity associated with gastrointestinal diseases, a promising alternative for treating this condition is utilizing molecules from natural products, given that their multi-target potential has been reported [23]. Natural products are excellent resources for identifying new pharmacological agents, given their structure and potential template for synthetic modification and optimization of selectivity and bioavailability [24]. Citral is a compound of plant origin found in lemon grass (*Cymbopogon citratus* and *C. flexuosus*), lemon balm (*Melissa officinalis*), and ginger (*Zingiber officinale*). It is widely used in the food, cosmetic, chemical, and pharmaceutical industries and is incorporated into fragrances, food, and beverages [25,26,27,28]. Chemically, citral is classified as an acyclic monoterpene, which is a mixture of two isomers: the neral cis isomer and geranial trans isomer [29]. Monoterpenes have been pharmacologically explored for their anti-inflammatory, antioxidant, and antibiotic activities. Considering gastric ulcers, monoterpenes were shown to exhibit activities ranging from injury prevention to accelerating the healing process [30]. Previously, we have reported that citral can promote gastroprotection in an NSAID-induced gastric injury model [31] however, the effect of citral on gastric ulcer healing in an obesity model remains unexplored. Therefore, we aimed to evaluate the effect of citral during the initial and late phases of the gastric ulcer healing process in eutrophic and obese mice.

## 2. Results

### 2.1. Induction of Obesity by HFD Ingestion

For the pharmacological evaluation of citral, the C57Bl/6 strain was used, given that this strain is more prone to the development of comorbidities resulting from obesity. Over 12 weeks (84 days), the body mass of each animal was measured twice weekly and from the third week onward. We detected a significant increase (10.1%; *p* < 0.05) in the body mass of HFD-fed animals when compared with that of SD-fed animals. Figure 1 shows the statistical difference from the third week after initiating HFD ingestion compared with animals fed an SD. At the end of the obesity induction period, the body mass of mice fed different diets differed by 51.8% (*p* < 0.0001), demonstrating that the employed strain responds well to the ingestion of an HFD, considering body weight gain. 

At the end of the treatment, animals were euthanized, and their abdominal (TAA), epididymal (TAE), and retroperitoneal (TAR) adipose tissues were harvested. To calculate the adiposity index, the masses of TAA, TAE, and TAR were summed, and the total value was divided by the mass of the animal. HFD-fed animals exhibited a 2.3-fold greater adiposity index (*p* < 0.0001) than those fed the SD (Table 1).

To verify the metabolic alterations, the lipid profile and the blood glucose levels of groups treated for 10 days were evaluated (Table 1). As shown in Table 1, the total cholesterol in HFD-fed animals was 31.8% (*p* < 0.001) higher than that in SD-fed animals. Likewise, the serum level of LDL was altered, with an increase of 1.8-fold (*p* < 0.0001) detected in HFD-fed animals when compared with that in SD-fed animals (Table 1). Despite the increase in total cholesterol and LDL, HDL levels were unchanged in the HFD-fed animals (Table 1). Therefore, we confirmed that at the end of 12 weeks of HFD ingestion, C57Bl/6 mice exhibited increased body mass, elevated adiposity index, and high serum levels of total and LDL cholesterol.

### 2.2. Acetic Acid-Induced Gastric Ulcer

Herein, we detected no regenerating tissue in the early phase of healing (Figure 2).

Accordingly, Figure 3 represents a comparison between treatments only in the regenerating tissue area of animals treated for 10 days after lesion induction. There was no significant difference in tissue regeneration areas between animals that received different treatments and the vehicle. 

We compared the base areas of the ulcer regardless of treatment and observed that HFD-fed animals exhibited a reduction in lesion area in 35.2% of the stomachs when compared with that in SD-fed (*p* < 0.001); this difference is shown in Figure 4B. Despite the difference in the initial phase, there was no significant difference in the area of injury between diets after 10 days of treatment, indicating that the healing rate in SD-fed animals was higher and faster than those in HFD-fed animals. 

Regarding the evolution of healing, all groups fed an SD showed a significant reduction in the ulcer base area over time. Among HFD-fed animals, only the groups that received daily treatment with 100 and 300 mg/kg citral exhibited a significant reduction in the ulcer base area between days 3 and 10 after ulcer induction. Figure 5 presents a representative image of one stomach from each experimental group.

### 2.3. MMP Activity

To more specifically evaluate the effects of citral, especially at doses exhibiting a significant reduction in the ulcer base area in HFD-fed animals, we quantified the activity of MMP-2 and -9 (n = 4–5), which are known to mediate ECM degradation, cellular migration, and re-epithelialization (Figure 6). Regarding temporal evolution, comparing only the different treatment periods regardless of the treatment received, we detected a 20% increase in MMP-2 activity in the stomachs of animals that received treatment for 10 days when compared with those treated for 3 days (*p* < 0.05). 

Figure 6A shows the rate of MMP-2 activation. The MMP-2 activation rate was significantly increased in all groups, except in HFD-fed animals administered 100 mg/kg citral and SD-fed animals administered 300 mg/kg citral. It should be noted that groups exhibiting no changes in MMP-2 activation rate demonstrated the highest rate of lesion area reduction.

Considering the MMP-9 activity, we detected no difference between treatments when comparing groups that received the same diet and those treated for the same period. In addition, diet change induced no significant difference within animal groups receiving the same treatment for the same duration. We evaluated the temporal evolution of MMP-9 activity and detected a difference between the initial and late phase MMP-9 activity in the SD-fed animals administered 25 mg/kg citral. Within the HFD-fed animals, the MMP-9 activity was significantly increased between days 3 and 10 in animals administered lansoprazole and 100 mg/kg citral (Figure 6B).

We speculate that the reduction in the ulcer base area could be associated with decreased MMP-9 activity, given that this enzyme acts mainly in the inflammatory phase of the lesion; consequently, reduced MMP-9 activity indicates the resolution of the lesion and its progression to the proliferative phase, during which its activity is more harmful than beneficial.

## 3. Discussion

Diet is an important determinant of human health and disease, and imbalanced eating habits are critical risk factors for obesity and metabolic disorders [4]. Ingestion of a HFD induces an increase in free fatty acids, responsible for raising the amount of reactive oxygen species, which defines oxidative stress and is responsible for the increased expression of pro-inflammatory cytokines, resulting in chronic low-grade inflammation [32]. The association between obesity and peptic ulcers has been studied, but the results of these investigations are still limited. A study carried out with health professionals in the United States of America showed an increased risk of gastric ulcers in people with obesity [33]. Therefore, the objective of the present study was to determine whether citral could overcome these changes in the healing of gastric ulcers induced by acetic acid. 

To examine the association between obesity and the ulcer healing profile, we selected the C57Bl/6 strain, as it has a greater predisposition for developing metabolic alterations associated with HFD ingestion [34]. Some studies have shown that the time of HFD ingestion can impact serum changes [4]. In the present study, we confirmed changes in body weight, adiposity index, and lipid profile (Table 1); therefore, the use of the C57Bl/6 strain proved to be efficient in meeting our objective. 

The main limitation of this study was the severity of the injury caused by acetic acid in the stomach of mice. In studies using rat models, lansoprazole, belonging to the class of proton pump inhibitors and the most frequently used pharmacological class for treating gastric ulcers, was found to be effective in reducing the lesion area of animals with acetic acid-induced ulcers [35]. However, in the present study, we did not detect a significant difference in the lesion area between lansoprazole- and vehicle-treated animals. Despite this limitation, we detected changes in the healing process promoted by other oral treatments administered. 

We evaluated 3- and 10-day treatment durations, given that these are established healing phases during which distinct types of mediators are known to act: three days after lesion induction, inflammatory cells are predominant, along with tissue necrosis and formation of granulation tissue in the ulcer margin. Until approximately day 10 after injury induction, intense migration of epithelial cells occurs. It leads to re-epithelialization of the ulcer and angiogenesis in the ulcer bed, at which point the completion phase of the healing process is initiated [36]. Herein, our data were corroborated by the significant differences in the area of regenerated tissue between animals treated for 3 and 10 days (Figure 4). 

The ECM has been widely explored as a pharmacological target, especially when tissue repair is required, as the dynamics of this complex structure can determine several crucial factors for the healing process [37]. Gastric lesions are directly associated with the degradation of the ECM, wherein MMPs play a crucial role [20]. Experimental and clinical studies have reported that the synthesis and activities of MMP-2 and -9 are altered during metabolic syndrome or according to variations in diet types ingested. Patients with obesity reportedly exhibit an increase in both MMP-2 and -9 plasma levels [22]. In animal models, MMP levels in adipose tissue differ according to the type of diet administered; a decrease in MMP-9 was reported in animals fed a diet rich in sucrose, with no change in plasma levels. However, in HFD-induced obese animals, increased MMP-9 activity has been reported in abdominal adipose tissue [17]. 

Regarding the healing of gastric ulcers, it can be postulated that appropriate healing will occur when activities of MMP-2 and -9 decrease gradually, given that in the late period, their actions are detrimental to the healing process [18]. In the present study, the reduction in the ulcer base area observed in the C2-HFD group could be associated with decreased MMP-9 activity, as this enzyme mainly acts during the inflammatory phase of the lesion. Consequently, the reduction in its activity indicates the resolution of the lesion and its progression to the proliferative phase, during which MMP-9 activity is more harmful than beneficial [18]. Furthermore, the presence of MMP-9 in chronic ulcers has been linked to lesion recurrence [20]; hence, in the late phase of healing, the reduction in its activity can be considered advantageous for the resolution of tissue injury. Herein, we analyzed MMP-2 bands separately and classified them as pro and active. Based on these values, we obtained the activation rate of these enzymes for each treatment. Notably, only animals in the SD-C3 and HFD-C2 groups did not exhibit a significant and gradual increase in MMP-2 activation. Considering this finding along with the lesion area of each treatment, we suggest that modulation of MMP-2 activity was achieved by treatments that did not significantly increase MMP-2 activation within 10 days of lesion induction, thereby potentially contributing to the reduced lesion area, as high MMP-2 activity in the late healing phase is indicative of poor healing.

Several studies in different models of gastric lesion induction show the importance of metalloproteinases in the course of healing, and the results obtained in this investigation provide information that can be used to deepen studies on the role of citral in the synthesis and degradation of each of the components of the extracellular matrix [38,39,40].

Thus, we demonstrated the effect of citral on temporal modulation of MMP-2 and -9 in wound healing in acetic acid-induced gastric ulcers in SD and HFD-fed mice. It is also known that the association between MMPs and collagen synthesis is an important factor in the development of wound healing. The collagen is the most abundant protein in ECM, which gives structural and functional support for cells, and its degradation is involved in tissue repair and wound healing. The denatured collagen is called gelatin and is degraded by gelatinases, mainly MMP-2 and 9. These MMPs also degrade some ECM glycoproteins and cytokines such as IL-8 and IL-1β [41].

Collagen deposition in ulcer bed is a determinant factor for healing while collagen peptides can promote cell migration and proliferation. In a study that evaluated the oral treatment with human-like collagen and human-like collagen products in rats, a reduction in lesion area was shown and these treatments enhanced cell and microvessel proliferation. Furthermore, the human-like collagen treatment promoted collagen deposition [15]. Moreover, an in vitro study has shown that citral treatment in osteoblast-like MG-63 cells induced an increase of collagen levels, leading to an osteogenic effect [42]. In addition to it, a study evaluating the effect of nano emulsion polyvinyl alcohol/chitosan hybrid incorporated with citral on healing of infected full-thickness skin wound showed an elevation in the amount of collagen and fibroblast infiltration in granulation tissue, accompanied by significant ulcer area reduction [43].

In present study, it is possible to hypothesize that citral acts in collagen availability in the gastric tissue, mainly due to the significant decrease in MMP-9 activity, the non-exacerbated elevation of MMP-2 activity in ulcers, decreasing the ulcer bed area after 10 days of treatment.

When evaluating the action of citral against gastric ulcer associated with obesity, we found that citral promotes different responses in animals that receive different diets. The initial hypothesis of this study was that the low-grade inflammatory condition caused by obesity would make it difficult to respond to pharmacological treatments, but the results obtained so far have shown the opposite. The variation in the pharmacological response in animals fed SD and HFD may be due to the increase in adipose tissue, which generates an increase in body weight and changes physiological functions such as blood flow distribution, modifying the pharmacokinetics of drugs in obese individuals [44]. There are some hypotheses to explain the different responses observed in the same treatment: the direct interaction of the treatment administered with the food ingested, which can occur due to physiological reactions, including changes in pH, gastrointestinal tract motility or secretion of bile acids. In addition, body composition also influences drug pharmacokinetics with regard to drug absorption, distribution, and metabolism. Some drugs are affected by food intake in general, and it is suggested that diets with a high calorie and lipid content have the greatest effect on pharmacokinetic properties [44]. Therefore, the results observed in the present study reiterate the importance of particularly evaluate the pharmacological activity in organisms with different nutritional conditions.

## 4. Materials and Methods

### 4.1. Animals

C57Bl/6 male mice, 120 fed SD and 120 fed HFD (total = 250 mice, 120 for each period evaluated), aged 4 to 6 weeks, from the Multidisciplinary Center for Biological Research in the Field of Science in Laboratory Animals (UNICAMP). All animals were acclimatized to the conditions of the sectorial vivarium for at least seven days before experimental manipulation, under a controlled temperature ranging between 23 and 25 °C and a 12-h light-dark cycle. The mice were housed in solid-bottom boxes lined with wood shavings and were fed a commercial diet (Presence, Jaguaribara, CE, Brazil) or high-fat diet (PragSoluções Biociências, Jaú, SP, Brazil) and water *ad libitum*. All protocols were approved by the Ethics Committee on the Use of Animals of the Institute of Biosciences of Botucatu (approval number 1208).

### 4.2. Obesity Induction

For obesity induction, animals were divided into two groups. The first group received a commercial diet (SD, Presence, Jaguaribara, CE, Brazil) and the second group received a high-fat diet (HFD, PragSoluções Biociências, Jaú, SP, Brazil). The HFD, which has 60% of its total calories from lipids, was prepared following the supplier’s instructions (PragSoluções Biociências, Jaú, SP, Brazil). All components, shown in Table 2, were weighed on a precision scale and mixed, respecting biosafety and then stored in 1 kg packages at 4 °C. Comparatively, the SD has only 16% of its total calories from lipids. The dietary intervention was performed for 12 weeks (84 days), and animals were weighed twice weekly, along with the food intake. Comparatively, the SD has only 16% of its total calories from lipids. The dietary intervention was performed for 12 weeks (84 days), and animals were weighed twice weekly, along with the food intake [44].

### 4.3. Experimental Design

We evaluated the citral (Sigma, St. Louis, MO, USA)-mediated effects during the early (3 days) and late phases (10 days) of healing. For both, mice were divided into two diet groups: standard diet (SD) and high-fat diet (HFD). After obesity induction, each diet group was redivided into six subgroups (n = 10/each) according to oral treatments: vehicle (V—Tween 80, 1%); lansoprazole (L—30 mg/kg); three doses of citral (C1—25 mg/kg; C2—100 mg/kg and C3—300 mg/kg) and sham, the subgroup that underwent laparotomy without lesion induction and was not treated (Figure 7). These divisions totalize 24 subgroups considering healing phases, diets and oral treatments. Citral doses were selected based on previous studies [31]. For inducing euthanasia, mice were anesthetized with isoflurane (Isoforine^®^, Cristália–Produtos Químicos Farmacêuticos Ltd.a., Itapira, SP, Brazil), and blood was obtained by cardiac puncture 24 h after administering the last treatment; subsequently, blood was centrifuged to collect the sera for further biochemical estimations.

### 4.4. Acetic Acid-Induced Gastric Ulcer 

For lesion induction, animals were anesthetized using isoflurane. The induction rate was 2%, and the maintenance rate was 2.5 to 3% in the low-flow inhalation anesthesia system equipment (Bonther, Ribeirão Preto, SP, Brazil). After anesthesia induction, the animals underwent laparotomy, and the stomach was exposed topically to 25 µL of acetic acid 80%. The contact area with acetic acid was limited using a plastic ring (diameter, 3.75 mm). After 20 s of exposure, acetic acid was removed, and the stomach was washed with saline (0.9% NaCl) to remove any residue. After establishing the lesion, the animals were sutured and returned to their respective boxes [45,46]. To evaluate the effects of citral, the animals were divided into five treatment groups: vehicle (V), Tween 80, 1%, used as a negative control; lansoprazole (L) (Ravoos Laboratories, Hyderabad, India), at a dose of 30 mg/kg, a drug used to treat gastric ulcers; citral (Sigma, St. Louis, MO, USA) at doses of 25, 100, and 300 mg/kg (C1, C2, and C3, respectively). For each experiment, the sham group included animals that underwent laparotomy without inducing injury. As the protocol involved a surgical procedure, a considerable loss of animals was observed during and after surgery until the experimental endpoint. Herein, we recorded a loss of approximately 25% in eutrophic and obese animals.

### 4.5. Biochemical Parameter

The serum levels of glucose, total cholesterol, low-density lipoprotein (LDL), high-density lipoprotein (HDL) and triglycerides were evaluated, using the colorimetric kits for glucose (Ref. K082 BioClin, Belo Horizonte, MG, Brazil), tryglicerides (Ref. K117, BioClin, Belo Horizonte, MG, Brazil), LDL (Ref. K088, BioClin, Belo Horizonte, MG, Brazil), HDL (Ref. F03120B, DIALAB Produktion und Vertrieb von chemisch-technischen Produkten und Laborinstrumenten, Neudorf, Austria) by dry chemistry according to the manufacturer’s instructions.

### 4.6. Macroscopic Analysis

To analyze the lesion area, we adopted the methodology previously described to identify ulcer structures [15]. The established methodology was modified to allow macroscopic aspect evaluation according to the following classification: the ulcer was considered the thinnest part surrounded by the regenerating tissue (Figure 2). Tumescent tissue around the ulcer was classified as regenerating tissue. Quantification was performed using the ImageJ software (National Institute of Mental Health, Bethesda, MD, USA).

### 4.7. Quantification of MMP-2 and MMP-9 by Zymography 

The lesion tissue was separated and homogenized with an extraction buffer (50 mM Tris-HCl pH 7.4, 0.2 M NaCl, 0.1% Triton-X 100, 10 mM CaCl2, and 1% protease inhibitor cocktail). The homogenates were subjected to 8% sodium dodecyl sulfate-polyacrylamide gel electrophoresis containing 1 mg/mL gelatin (Sigma, St. Louis, MO, USA) under non-reducing conditions. The gels were washed twice in 2.5% Triton-X-100 (Sigma, St. Louis, MO, USA) and then incubated in calcium assay buffer (40 mM Tris and HCl, pH 7.4, 0.2M NaCl, and 10 mM CaCl2) for 18 h at 37 °C. The gels were stained with 0.1% Coomassie blue, followed by decolorization. Zones with gelatinolytic activity served as negative staining. The zymographic bands were quantified using optical densitometry linked to ImageJ software (National Institute of Mental Health, Bethesda, MD, USA) [46].

### 4.8. Statistical Analysis

Data are expressed as mean ± standard error of the mean (S.E.M.) of examined parameters. A one-way analysis of variance (ANOVA) followed by the Student’s t-test was performed to compare the effect of diets, regardless of the treatment administered. To evaluate the temporal evolution of healing, different periods of the same treatment were compared between animals receiving the same diet; for this comparison, a two-way ANOVA followed by the Bonferroni test was performed. To compare the treatment effects within groups receiving the same diet, the Dunnet test was performed, in which comparisons were made between groups that received lansoprazole and citral at three doses relative to the negative control group. Comparisons were separately performed for the 3- and 10-day treatments. Finally, to verify the effects of different diets, we compared the evolution of lesions within the same treatment administered for the same period using the Bonferroni post-test. For all analyses, *p* < 0.05 was accepted as statistically significant. All experimental dates were analyzed using GraphPad Prism (GraphPad Software, San Diego, CA, USA).

## 5. Conclusions

Herein, we demonstrated that 100 mg/kg citral administration could enhance the healing rate in obese C57Bl/6 mice when compared with the other treatments and diet ingestion. This improvement of the reduced lesion area was accompanied by decreased tissue MMP-9 and modulation of MMP-2 activation.

## Figures and Tables

**Figure 1 ijms-24-04888-f001:**
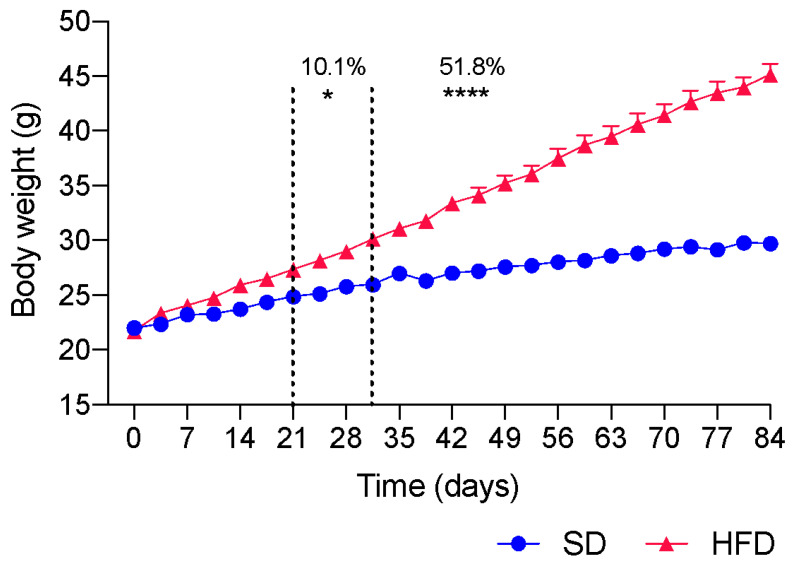
Comparison of body weight evolution between animals fed standard diet (SD) and high-fat diet (HFD) throughout the period of obesity induction. Results are expressed as mean ± s.e.m (n = 96–106). Statistical significance was determined by two-way ANOVA followed by Bonferroni’s test * *p* < 0.05 and **** *p* < 0.0001.

**Figure 2 ijms-24-04888-f002:**
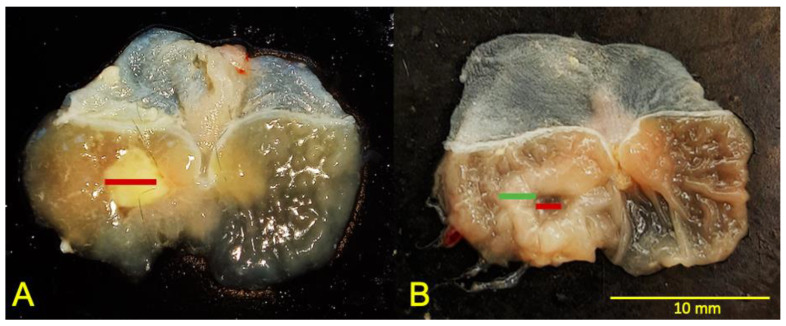
Representative images of stomachs of animals treated for 3 (**A**) and 10 (**B**) days. The red lines represent the base of the ulcer and the green line the regenerating tissue.

**Figure 3 ijms-24-04888-f003:**
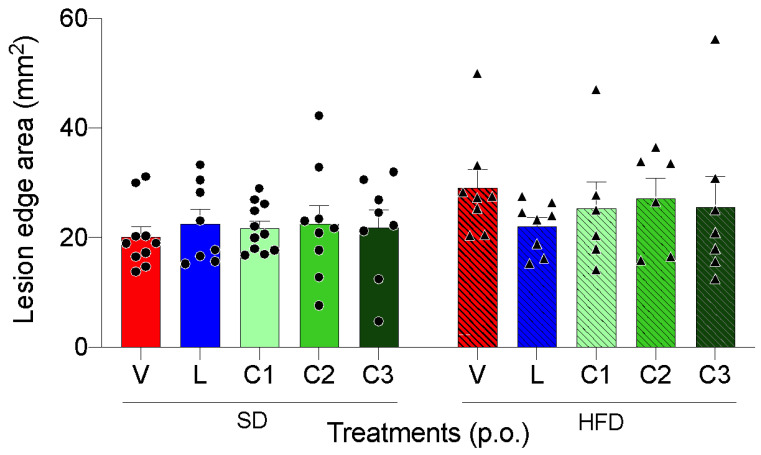
Lesion edge area in animals fed SD (standard diet) and HFD (high-fat diet) and treated with V—Vehicle, L—Lansoprazole, C1—Citral (25 mg/kg), C2—Citral (100 mg/kg kg) and C3—Citral (300 mg/kg). Circles and triangles represent individual values of SD and HFD-fed animals, respectively. Results are expressed as mean ± s.e.m (n = 6–11) and statistical significance was determined by two-way ANOVA followed by Tukey’s test.

**Figure 4 ijms-24-04888-f004:**
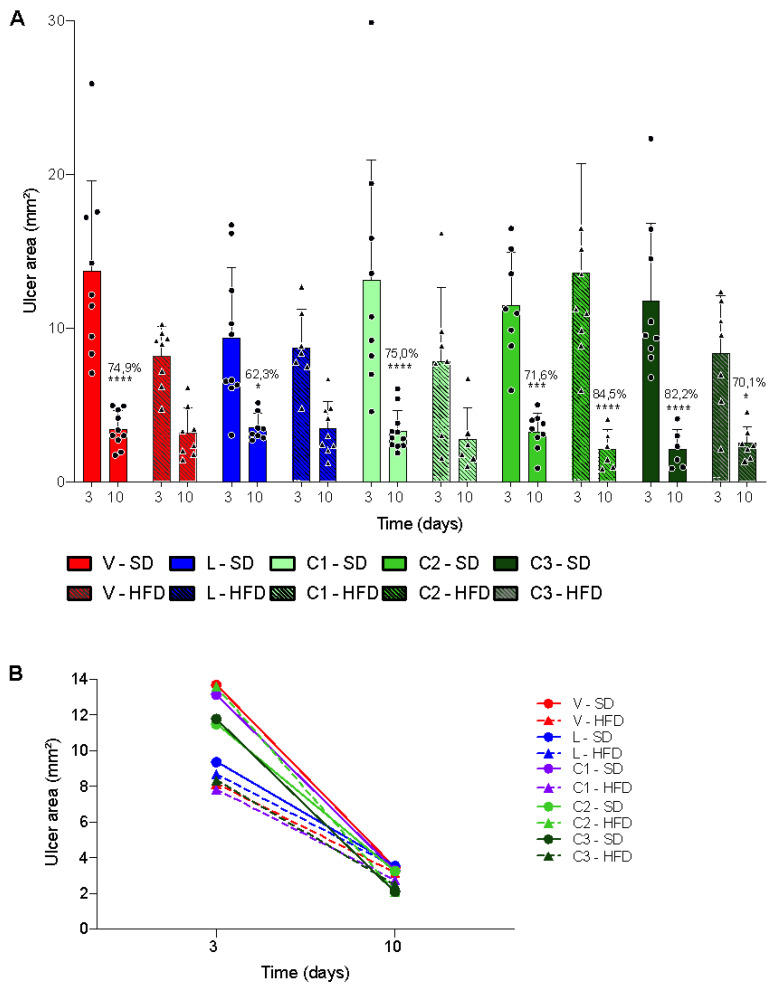
Temporal evolution of the ulcer area in animals fed with SD (standard diet) and HFD (high-fat diet). (**A**) Individual values of each group. The lesion areas were compared on the tenth day in relation to the third day after induction of lesion of the animals treated with V—Vehicle, L—Lansoprazole, C1—Citral (25 mg/kg), C2—Citral (100 mg/kg) and C3—Citral (300 mg/kg). Circles and triangles represent individual values of SD and HFD-fed animals, respectively. Results are expressed as mean ± s.e.m (n = 6–11) and statistical significance was determined by two-way ANOVA followed by Bonferroni’s test, with * *p* < 0.05; *** *p* < 0.001 and **** *p* < 0.0001. (**B**) Comparison of healing evolution between groups over time.

**Figure 5 ijms-24-04888-f005:**
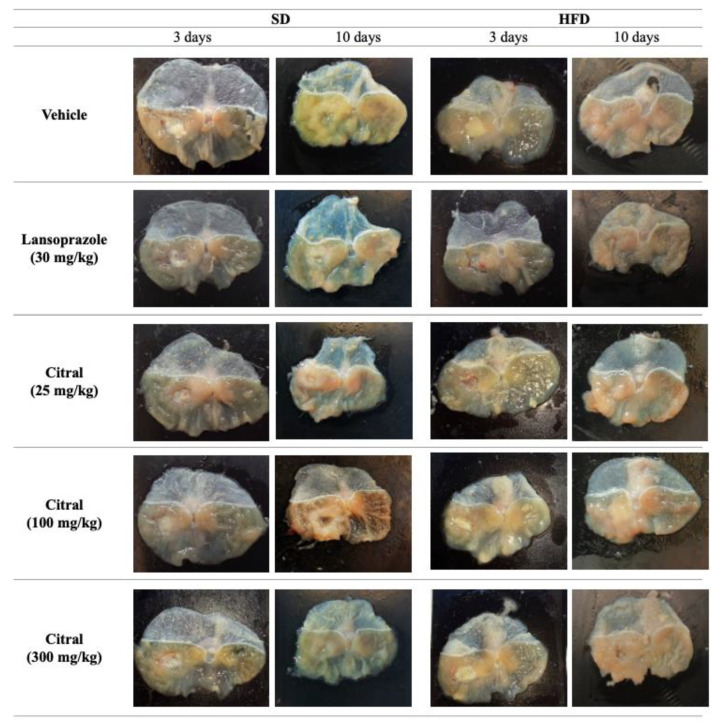
Macroscopic images of mice stomach with acetic acid-induced lesion after three and ten days of treatment.

**Figure 6 ijms-24-04888-f006:**
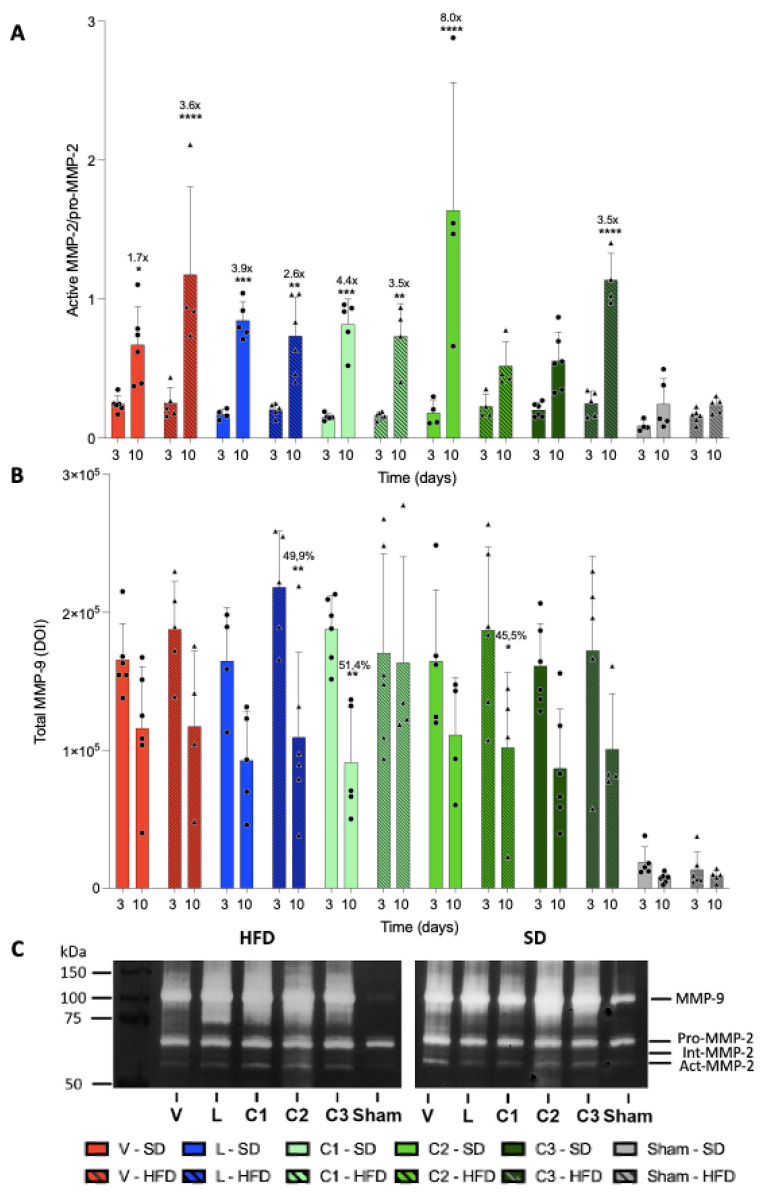
Evaluation of the activity of MMP-2 and 9 by zymography. (**A**) Active MMP-2/Pro-MMP-2 rate in the lesions of animals fed SD (standard diet) and HFD (high-fat diet). (**B**) Temporal evolution of MMP-9 activity in the lesions of animals fed SD (standard diet) and HFD (high-fat diet). TMMP-2 and 9 activities were compared on the tenth day in relation to the third day after induction of injury of animals treated with V—Vehicle, L—Lansoprazole, C1—Citral (25 mg/kg), C2—Citral (100 mg/kg) and C3—Citral (300 mg/kg). Circles and triangles represent individual values of SD and HFD-fed animals, respectively. Results are expressed as mean ± s.e.m (n = 4–6) and statistical significance was determined by two-way ANOVA followed by the Bonferroni test, with * *p* < 0.05; ** *p* < 0.01; *** *p* < 0.001 and **** *p* < 0.0001. (**C**) Images of gels with the bands of gelatinolytic activity of MMPs in the zymography technique. On the left, the weights of the molecular weight marker and, on the right, the discrimination of the bands obtained in the gels.

**Figure 7 ijms-24-04888-f007:**
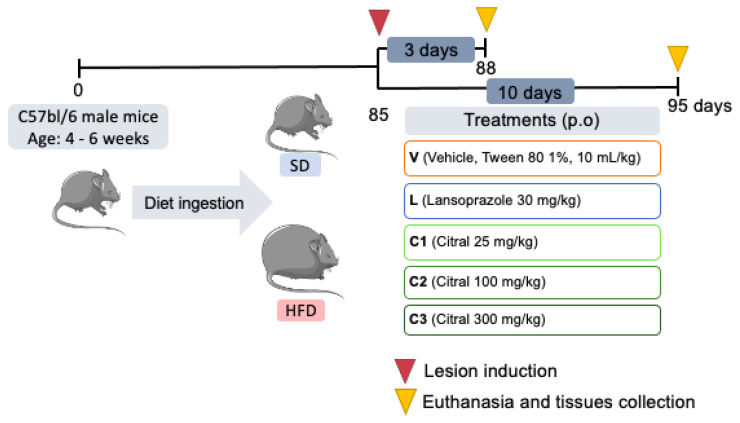
Experimental design to evaluate citral in the early (3 days) and late (10 days) phases of gastric ulcer healing in C57Bl/6 eutrophic and obese male mice.

**Table 1 ijms-24-04888-t001:** Biochemical parameters of mice after induction of gastric ulcer by acetic acid and 10 days of treatment. Data are represented as mean ± s.e.m. SD = standard diet and HFD = high-fat diet.

	SD					
	Vehicle 10 mL/kg	Lansoprazole 30 mg/kg	Citral 25 mg/kg	Citral 100 mg/kg	Citral 300 mg/kg	Sham
Adiposity index (%)	2.34 ± 0.09 ^a^	2.03 ± 0.16 ^a^	2.48 ± 0.33 ^a^	2.22 ± 0.19 ^a^	1.99 ± 0.20 ^a^	2.76 ± 0.15 ^a^
Glucose (mg/dL)	204.5 ± 12.85	203.2 ± 15.96	184.50 ± 5.69	205.66 ± 7.67	180.33 ± 15.93	167.4 ± 9.89
Total cholesterol (mg/dL)	136.7 ± 5.59 ^a^	133.00 ± 6.53 ^a^	140.84 ± 4.63 ^a^	135.17 ± 5.31 ^a^	128.00 ± 7.49 ^a^	138.80 ± 5.57 ^a^
HDL (mg/dL)	78.0 ± 5.08 ^a^	83.34 ± 3.92 ^a^	77.17 ± 3.14 ^a^	78.34 ± 3.28 ^a^	80.67 ± 3.04 ^a^	75.40 ±5.50 ^a^
LDL (mg/dL)	28.50 ± 8.17	22.84 ± 5.21	35.67 ± 6.07	34.34 ± 6.60	22.34 ± 4.78	35.20 ± 8.96
Triglycerides (mg/dL)	151.0 ± 16.95	134.00 ± 14.89	142.30 ± 20.12	112.00 ± 12.70	124.67 ± 17.13	142.40 ± 15.86
	HFD					
Adiposity index (%)	7.51 ± 0.62 ^b^	8.64 ± 0.35 ^b^	6.92 ± 0.92 ^b^	7.33 ± 0.52 ^b^	7.24± 0.66 ^b^	8.84 ± 0.47 ^b^
Glucose (mg/dL)	196.83 ± 30.10	280.30 ± 21.27	179.20 ± 42.91	205.20 ± 22.17	178.33 ± 22.23	242.40 ± 27.00
Total cholesterol (mg/dL)	184.00 ± 27.59 ^b^	221.80 ± 25.32 ^b^	156.20 ± 19.12 ^b^	150.30 ± 20.32 ^b^	185.70 ± 16.47 ^b^	202.80 ± 12.15 ^b^
HDL (mg/dL)	75.00 ± 5.30 ^b^	76.34 ± 4.64 ^b^	65.60 ± 11.42 ^b^	72.17 ± 3.12 ^b^	75.67 ± 3.56 ^b^	73.00 ± 2.55 ^b^
LDL (mg/dL)	87.67 ± 24.42	125.30 ± 23.55	75.00 ± 13.75	60.34 ± 17.13	91.50 ± 15.65	91.80 ± 23.52
Triglycerides (mg/dL)	107.00 ± 22.12	100.80 ± 15.37	78.00 ± 7.18	88.67 ± 10.80	92.34 ± 8.29	90.80 ± 4.77

Data are represented as mean ± s.e.m. SD = standard diet and HFD = high-fat diet. Different letters represent difference between diets.

**Table 2 ijms-24-04888-t002:** Composition of the high fat diet (HFD).

Component	g/kg	kcal/kg
Corn starch (q.s.)	115.5	462
Casein	200	800
Sucrose	100	400
Maltodextrin	132	528
Lard	312	2808
Soy oil	40	360
Cellulose	50	-
Mineral mix	35	-
Vitamin mix	10	-
L-Cystine	3	-
Choline	2.5	-
Total	1000	5358

## Data Availability

The data that support the findings of this study are available from the corresponding author, R.O., upon reasonable request.

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
