# Peer review of "Citral Modulates MMP-2 and MMP-9 Activities on Healing of Gastric Ulcers Associated with High-Fat Diet-Induced Obesity"

_ijms, 2023, doi:10.3390/ijms24054888_

Round 1
Reviewer 1 Report
The major comments are as follows:
It is pretty standard to observe higher weight gain after feeding mice with high-fat diet. I fail to understand the significance of the figure 1. It is such a standard thing in the field that you don’t need to establish that c57bl6 mice gain weight on HFD (page 3 line 137-138). Similarly, the adiposity index data could be used as a supplemental figure to show that 10 days of treatment does not affect the adiposity, which is not the main findings of the paper anyway. With so much of trivial and excess information, the actual finding gets somewhat lost in unnecessary details.
The experimental groups need to be better explained. It is not at all clear whether all the mice on HFD went through acetic acid-mediated gastric ulcer induction. ‘Sham-SD: animals fed an HFD and underwent laparotomy without lesion induction’ page 13, line no. 358-359. I am guessing these should have been ‘Sham-HFD’. However, I don’t see any data from Sham SD and Sham HFD mice for figures 3-9. It is hard to interpret the data if we are missing out on the sham and the baseline effects of L and C. It is important to include the sham SD and sham HFD data for table 1 too. Furthermore, it should be noted that with highly variable results, which is quite understandable in mice on an HFD, the experiment should be repeated three independent times to conclude on the effects of a drug. I believe this data is preliminary and intriguing and warrants further rigorous experimental assessment.
The data provided here is not sufficient to claim that citral works through MMP-2 and 9 regulations. There are several parameters of collagen biosynthesis, collagen deposition and secretion, and inflammatatory signals that needs careful assessment in vivo. I believe that the paper will improve if the authors focus on the abovementioned parameters, and in addition, assess in vitro whether citral affects MMPs and collagen in fibroblasts at baseline and after induction with inflammatory and anti-inflammatory cytokines, and whether citral affects collagen degradation. Furthermore, the paper will benefit from assessing the half-life and bioavailability of citral in mice to get some insights to the much-needed pharmacokinetics.
Author Response
(x) Moderate English changes required
Answer: Thank you very much for your attention in reviewing this manuscript. All the English text of this work was revised and corrected by the company Editage.
Point 1: It is pretty standard to observe higher weight gain after feeding mice with high-fat diet. I fail to understand the significance of the figure 1. It is such a standard thing in the field that you don’t need to establish that c57bl6 mice gain weight on HFD (page 3 line 137-138). Similarly, the adiposity index data could be used as a supplemental figure to show that 10 days of treatment does not affect the adiposity, which is not the main findings of the paper anyway. With so much of trivial and excess information, the actual finding gets somewhat lost in unnecessary details.
Answer 1: Thank you for taking the time to review our manuscript. Obesity induction results are relevant information to show the obese animals in the present study. Thus, adiposity index data was added to Table 1 as descriptions of the obesity profile of the animals used in this study. In previous studies, our group showed the same profile of obese animals in the table (doi: https://doi.org/10.3390/biom10101454) as a way to ensure the acquisition of both obese and lean animals for the evaluation of the present study. Considering your suggestions, we kept the evolution of body mass in a graph, and the other data such as adiposity index and lipid profile are in table form in order to reduce the excess of information.
Point 2: The experimental groups need to be better explained. It is not at all clear whether all the mice on HFD went through acetic acid-mediated gastric ulcer induction. ‘Sham-SD: animals fed an HFD and underwent laparotomy without lesion induction’ page 13, line no. 358-359. I am guessing these should have been ‘Sham-HFD’.
Answer 2: Thank you for your attention. The text has been changed according to the suggested corrections.
Point 3: However, I don’t see any data from Sham SD and Sham HFD mice for figures 3-9. It is hard to interpret the data if we are missing out on the sham and the baseline effects of L and C. It is important to include the sham SD and sham HFD data for table 1 too.
Answer 3: I appreciate your consideration. As the animals in the Sham group did not undergo injury induction, the lesion area and the lesion border area, which are equal to zero, were not quantified. The quantifications of MMP-2 and 9 from the Sham group are shown in figure 6. As suggested, sham group data was also included in table 1.
Point 4: Furthermore, it should be noted that with highly variable results, which is quite understandable in mice on an HFD, the experiment should be repeated three independent times to conclude on the effects of a drug. I believe this data is preliminary and intriguing and warrants further rigorous experimental assessment.
Answer 4: I thank you for your consideration. We understand that the result is preliminary and we have made the necessary changes to the text to clarify this. On the other hand, we can highlight the difference in the activity of the citral effect in animals that receive different diets, which will serve as a background for future studies involving the treatment of diseases associated with obesity.
Point 5: The data provided here is not sufficient to claim that citral works through MMP-2 and 9 regulations. There are several parameters of collagen biosynthesis, collagen deposition and secretion, and inflammatory signals that need careful assessment in vivo. I believe that the paper will improve if the authors focus on the above mentioned parameters, and in addition, assess in vitro whether citral affects MMPs and collagen in fibroblasts at baseline and after induction with inflammatory and anti-inflammatory cytokines, and whether citral affects collagen degradation. Furthermore, the paper will benefit from assessing the half-life and bioavailability of citral in mice to get some insights to the much-needed pharmacokinetics.
Answer 5: I thank you for your consideration. Unfortunately, it is not possible to quantify other analytes with the same samples, as the sample size was limited. However, knowing the response profile and activity of matrix metalloproteinase (MMP) 2 and 9 may indicate an important mechanism of healing and tissue remodeling after damage during the gastric ulcer. Previous studies described that the total activity of the MMPs was higher in the initial phases of healing, MMP-2 has a very stable activity compared to the total MMP, while MMP-9 followed the pattern of the total. Thus, it is higher during the beginning of healing, concluding that the different MMP's have different actions within the healing process, with MMP-2 present mainly during prolonged healing, and MMP9 related to the wound epithelialization process, and events that precede the repair (DOI: 10.1111/j.1365-2133.1994.tb04974.x). In rats, the MMPs activity was described in the models of acetic acid-induced ulcer gastric that showed the important role of these enzymes in removing damaged tissue (clearing) and helping repair damage (DOI: https://doi.org/10.3109/00365529709025075). Another study shows the temporal role of MMP-2 and MMP-9 during the indomethacin-induced gastric ulcer as important factors in the repair process development (DOI: https://doi.org/10.1159/000008759). With information on the response of metalloproteinase activity to citral treatment, this study provides material that enables deeper investigations into the synthesis and degradation of matrix metalloproteinases. (p11, line 708).
Reviewer 2 Report
General Comments
The manuscript is well-written with broad referencing. The authors investigated “Citral, a monoterpene, modulate of MMP-2 and MMP-9 activities on healing of gastric ulcers associated with high-fat diet-induced obesity”. The study is interesting and adds to the existing body of knowledge. There are a few things that need clarification and revisions. All details and comments are listed below.
Details Comments
1. Page 1, Lines 29-33: Please add citations
2. Page 1, Lines 35-38: Please add citations
3. Page 2, Lines 53-56: Please add citations
4. Page 3, Lines 115-117: Please add citations for each species of the plant reported in this compound (Citral).
5. Page 3, Lines 130-138: Did the authors measure the food intake in the present study?
6. Page 4, Lines 156-157: Authors showed mention which groups they are comparing. In Figure 3A, there are groups V, L, C1, C2, and C3 for animals fed with standard diet and High-fat diet. For example, subgroups of SD (V) vs. subgroups of HFD (V) showed p<0.001.
7. Page 5, Lines 158-159: Please see the above comments. Describe in detail.
8. Page 5, Lines 164-165: Table legend. Please remove “Data are represented as mean ± s.e.m. SD= standard diet and HFD= high-fat diet” and put it in the Table footer. At the Table footer, please add sample size (n= ?) and significant value ( p < 0.05) for each group that showed a significant difference.
9. Page 6, Line 167: No graph for Triglycerides levels?
10. Page 9, Lines 206-207: In the Figure legend, Please add sample size (n = ?)
11. Page 10, Line 225: Please combine Figure 8 and Figure 9.
12. Page 13, Lines 334-339: Please provide the animal ethics approval number
13. Page 13, Line 334: Please provide the total of rats used in the whole study period and add the body weight range of rats.
14. Page 13, Line 346: High-fat diet; the diet was prepared by authors (investigators) or purchased from where? Please provide the company name. If prepared, please explain the details of the preparation and add citations.
15. Page 14, Lines 369-371: Please add citations for these protocols.
16. Page 14, Lines 386-389: Please add the catalog number for each kit (TC, TG, LDL, HDL)
17. Page 14, Line 391: ..described by Xing et al. [13] to identify ulcer structure.
18. Page 15, Lines 410-421: What software did the author use to analyze data? (eg. SPSS, GraphPad Prism..)
19. Page 15, Lines 440-442: Please also put this animal's ethical approval number in Section 4.1.
Author Response
General Comments
The manuscript is well-written with broad referencing. The authors investigated “Citral, a monoterpene, modulate of MMP-2 and MMP-9 activities on healing of gastric ulcers associated with high-fat diet-induced obesity”. The study is interesting and adds to the existing body of knowledge. There are a few things that need clarification and revisions. All details and comments are listed below.
Details Comments
- Page 1, Lines 29-33: Please add citations
- Page 1, Lines 35-38: Please add citations
- Page 2, Lines 53-56: Please add citations
- Page 3, Lines 115-117: Please add citations for each species of the plant reported in this compound (Citral).
Answer: Thank you for your attention, we have added the respective references (points 1, 2, 3, and 4) as indicated in each line of the manuscript.
- Page 3, Lines 130-138: Did the authors measure the food intake in the present study?
Answer 5: I appreciate this consideration, food intake was measured in previous studies to validate the methodology. In this study, intake was measured only for experimental control purposes.
- Page 4, Lines 156-157: Authors showed mention which groups they are comparing. In Figure 3A, there are groups V, L, C1, C2, and C3 for animals fed with standard diet and High-fat diet. For example, subgroups of SD (V) vs. subgroups of HFD (V) showed p<0.001.
Answer 6: Thank you for your attention in evaluating this manuscript. There was no difference between the groups that received different treatments, we only observed significant differences between HFD and SD animals with regard to the lipid profile. The results are described in table 1, as suggested by the first reviewer.
- Page 5, Lines 158-159: Please see the above comments. Describe in detail.
Answer 7: Thank you for your attention, the text has been corrected following the suggestions.
- Page 5, Lines 164-165: Table legend. Please remove “Data are represented as mean ± s.e.m. SD= standard diet and HFD= high-fat diet” and put it in the Table footer. At the Table footer, please add sample size (n= ?) and significant value (p < 0.05) for each group that showed a significant difference.
Answer 8: Thank you for this consideration, the text has been corrected following the referee’s suggestions (p5, line 208).
- Page 6, Line 167: No graph for Triglycerides levels?
Answer 9: Thank you for this consideration, lipid profile data are presented only in table form, following the instructions of the first reviewer.
- Page 9, Lines 206-207: In the Figure legend, Please add sample size (n = ?)
Answer 10: Thank you for your consideration, the sample size is described in figures 5 and 6, which show the results of measuring the areas of the edge and base of the lesion.
- Page 10, Line 225: Please combine Figure 8 and Figure 9.
Answer 11: Thank you for your attention, we add that we make the modifications to the indicated figures. In addition, we modified figure 7, as shown in Figure 6.
- Page 13, Lines 334-339: Please provide the animal ethics approval number
Answer 12: I thank you for consideration in our manuscript, we also add the animal's ethics approval number in section 4.1. Animals.
- Page 13, Line 334: Please provide the total of rats used in the whole study period and add the body weight range of rats.
Answer 13: I appreciate your consideration, this information was included in the text for clarity.
- Page 13, Line 346: High-fat diet; the diet was prepared by authors (investigators) or purchased from where? Please provide the company name. If prepared, please explain the details of the preparation and add citations.
Answer 14: Thank you for your consideration of our manuscript. We prepared the diet with the inputs provided by the PragSoluções company, as described in the manuscript (p13, line 512).
- Page 14, Lines 369-371: Please add citations for these protocols.
Answer 15: Thank you for your consideration in relation to this item from our manuscript. We added the references of protocols (p14, line 579), such as highlight in the text.
- Page 14, Lines 386-389: Please add the catalog number for each kit (TC, TG, LDL, HDL)
Answer 16: I appreciate your consideration, this information was included in the text for clarity.
- Page 14, Line 391: ..described by Xing et al. [13] to identify ulcer structure.
Answer 17: I thank you for the review in this manuscript, we modified the reference structure as highlighted in the manuscript.
- Page 15, Lines 410-421: What software did the author use to analyze data? (eg. SPSS, GraphPad Prism..)
Answer 18: I thank you for your attention, we have added the information from the GraphPad Prism software used in this work, as demonstrated in section 4.9. Statistical analysis (p15, line 639).
- Page 15, Lines 440-442: Please also put this animal's ethical approval number in Section 4.1.
Answer 19: I thank you for your consideration in our manuscript, we also added the animal's ethic approval number in section 4.1. Animals (p12, line 493).
Round 2
Reviewer 1 Report
I see here that the authors refrained from adding new experiments and addressing points raised in my previous review report point 4 and 5. The authors have not addressed the point 5 at all, that would add some new insights to the manuscript. Overall, the manuscript still presents very preliminary findings. Addressing some of the points raised in point 5 (especially related to collagen), would help increase the quality of the manuscript.
Author Response
Point 1: I see here that the authors refrained from adding new experiments and addressing points raised in my previous review report point 4 and 5. The authors have not addressed the point 5 at all, that would add some new insights to the manuscript. Overall, the manuscript still presents very preliminary findings. Addressing some of the points raised in point 5 (especially related to collagen), would help increase the quality of the manuscript.
Pontos 4 e 5:
Furthermore, it should be noted that with highly variable results, which is quite understandable in mice on an HFD, the experiment should be repeated three independent times to conclude on the effects of a drug. I believe this data is preliminary and intriguing and warrants further rigorous experimental assessment.
The data provided here is not sufficient to claim that citral works through MMP-2 and 9 regulations. There are several parameters of collagen biosynthesis, collagen deposition and secretion, and inflammatory signals that need careful assessment in vivo. I believe that the paper will improve if the authors focus on the above mentioned parameters, and in addition, assess in vitro whether citral affects MMPs and collagen in fibroblasts at baseline and after induction with inflammatory and anti-inflammatory cytokines, and whether citral affects collagen degradation. Furthermore, the paper will benefit from assessing the half-life and bioavailability of citral in mice to get some insights to the much-needed pharmacokinetics.
Answer:
I appreciate your contribution and thank you again for your time. Different studies have already demonstrated the MMP temporal action, which works together with the components of the extracellular matrix (ECM) to promote and modulate gastric tissue healing. The expression and deposition of collagen in the ulcer bed are determining factors for the remodelation of the tissue, while collagen peptides also can promote cell migration and proliferation. Thus, the modulation of collagen deposition through the activities of MMPs acts as a crucial factor during ulcer formation and healing (DOI: 10.1002/jemt.1108; DOI: 10.3109/00365529709025075). In our study, we showed the capacity of the citral to induce a significant decrease in MMP-9 activity, the non-exacerbated elevation of MMP-2 activity in ulcers, decreasing the ulcer bed area after 10 days of treatment. A in vitro study demonstrated that citral acts as a potential treatment that raises collagen synthesis in osteoblast-like MG-63 cells (DOI: 10.4103/pm.pm_242_20). Another study, evaluating the effect of nanoemulsion polyvinyl alcohol/chitosan hybrid incorporated with citral on healing of infected full-thickness skin wound in mice, showed an elevation in the amount of collagen and fibroblast infiltration in granulation tissue accompanied by significant ulcer area reduction (DOI: 10.1016/j.jddst.2022.103589). However, additional studies of collagen quantification after citral administration is necessary to validate our hypothesis.
Considering this data, we added a paragraph (p 11 line 769) with this information, explaining the relationship between collagen synthesis and the activity of metalloproteinases, evaluated in this study.